# Vascular Remodeling in Moyamoya Angiopathy: From Peripheral Blood Mononuclear Cells to Endothelial Cells

**DOI:** 10.3390/ijms21165763

**Published:** 2020-08-11

**Authors:** Francesca Tinelli, Sara Nava, Francesco Arioli, Gloria Bedini, Emma Scelzo, Daniela Lisini, Giuseppe Faragò, Andrea Gioppo, Elisa F. Ciceri, Francesco Acerbi, Paolo Ferroli, Ignazio G. Vetrano, Silvia Esposito, Veronica Saletti, Chiara Pantaleoni, Federica Zibordi, Nardo Nardocci, Maria Luisa Zedde, Alessandro Pezzini, Vincenzo Di Lazzaro, Fioravante Capone, Maria Luisa Dell’Acqua, Peter Vajkoczy, Elisabeth Tournier-Lasserve, Eugenio A. Parati, Anna Bersano, Laura Gatti

**Affiliations:** 1Laboratory of Cellular Neurobiology, Neurology IX Unit, UCV, Fondazione IRCCS Istituto Neurologico Carlo Besta, 20133 Milan, Italy; francesca.tinelli@istituto-besta.it (F.T.); sara.nava@istituto-besta.it (S.N.); francesco.arioli@studenti.units.it (F.A.); daniela.lisini@istituto-besta.it (D.L.); 2Immunology and Cell Therapy Unit, Tettamanti Research Center, University of Milano-Bicocca, 20900 Monza, Italy; gloriabedini85@gmail.com; 3Neurology IX Unit, UCV, Fondazione IRCCS Istituto Neurologico Carlo Besta, 20133 Milan, Italy; emma.scelzo@istituto-besta.it (E.S.); eugenio.parati@istituto-besta.it (E.A.P.); anna.bersano@istituto-besta.it (A.B.); 4Diagnostic Imaging & Interventional Neuroradiology Unit, Fondazione IRCCS Istituto Neurologico Carlo Besta, 20133 Milan, Italy; giuseppe.farago@istituto-besta.it (G.F.); andrea.gioppo@istituto-besta.it (A.G.); elisa.ciceri@istituto-besta.it (E.F.C.); 5Neurosurgical II Unit, Fondazione IRCCS Istituto Neurologico Carlo Besta, 20133 Milan, Italy; francesco.acerbi@istituto-besta.it (F.A.); paolo.ferroli@istituto-besta.it (P.F.); ignazio.vetrano@istituto-besta.it (I.G.V.); 6Developmental Neurology Unit, Fondazione IRCCS Istituto Neurologico Carlo Besta, 20133 Milan, Italy; silvia.esposito@istituto-besta.it (S.E.); veronica.saletti@istituto-besta.it (V.S.); chiara.pantaleoni@istituto-besta.it (C.P.); 7Department of Pediatric Neuroscience, Fondazione IRCCS Istituto Neurologico Carlo Besta, 20133 Milan, Italy; federica.zibordi@istituto-besta.it (F.Z.); nardo.nardocci@istituto-besta.it (N.N.); 8Neurology Unit, Stroke Unit, Azienda Unità Sanitaria Locale—IRCCS di Reggio Emilia, 42122 Reggio Emilia, Italy; Marialuisa.Zedde@ausl.re.it; 9Department of Clinical and Experimental Sciences, Clinica Neurologica, Università degli Studi di Brescia, 25121 Brescia, Italy; alessandro.pezzini@unibs.it; 10Unit of Neurology, Neurophysiology, Neurobiology, Department of Medicine, Università Campus Bio-Medico di Roma, 00128 Rome, Italy; v.dilazzaro@unicampus.it (V.D.L.); f.capone@unicampus.it (F.C.); 11Stroke Unit, Neurology Clinic, Department of Biomedical Metabolic and Neural Sciences, Nuovo Ospedale Civile S Agostino Estense, University of Modena and Reggio Emilia, 41126 Modena, Italy; dellacqua.marialuisa@aou.mo.it; 12Department of Neurosurgery, Charite Universitätsmedizin, 10117 Berlin, Germany; peter.vajkoczy@charite.de; 13Department of Genetics, Lariboisière Hospital and INSERM U1141, Paris-Diderot University, 75010 Paris, France; tournier-lasserve@univ-paris-diderot.fr

**Keywords:** neovascularization, Moyamoya angiopathy, endothelial progenitor cells, RNF213

## Abstract

The pathophysiological mechanisms of Moyamoya angiopathy (MA), which is a rare cerebrovascular condition characterized by recurrent ischemic/hemorrhagic strokes, are still largely unknown. An imbalance of vasculogenic/angiogenic mechanisms has been proposed as one possible disease aspect. Circulating endothelial progenitor cells (cEPCs) have been hypothesized to contribute to vascular remodeling of MA, but it remains unclear whether they might be considered a disease effect or have a role in disease pathogenesis. The aim of the present study was to provide a morphological, phenotypical, and functional characterization of the cEPCs from MA patients to uncover their role in the disease pathophysiology. cEPCs were identified from whole blood as CD45^dim^CD34^+^CD133^+^ mononuclear cells. Morphological, biochemical, and functional assays were performed to characterize cEPCs. A significant reduced level of cEPCs was found in blood samples collected from a homogeneous group of adult (mean age 46.86 ± 11.7; 86.36% females), Caucasian, non-operated MA patients with respect to healthy donors (HD; *p* = 0.032). Since no difference in cEPC characteristics and functionality was observed between MA patients and HD, a defective recruitment mechanism could be involved in the disease pathophysiology. Collectively, our results suggest that cEPC level more than endothelial progenitor cell (EPC) functionality seems to be a potential marker of MA. The validation of our results on a larger population and the correlation with clinical data as well as the use of more complex cellular model could help our understanding of EPC role in MA pathophysiology.

## 1. Introduction

Moyamoya angiopathy (MA) is a rare, chronic, and disabling cerebrovascular disease with a prevalence of 0.086–10.5/100,000. It is characterized by a progressive steno-occlusive lesion of the terminal part of the internal carotid arteries (ICAs) and their proximal branches, associated with the compensatory development of an unstable network of collateral vessels at the base of the brain (Moyamoya vessels) [1,2]. These vascular hallmarks are responsible for recurrent ischemic and hemorrhagic strokes (about 80% of cases), leading children and adults affected by MA to severe neurological (sensorimotor, speech, and cognitive) deficits, progressive physical disabilities, and even death [3,4]. MA is frequent in East Asian countries (i.e., 0.34–0.94/100,000 in Japan) while rarely reported in Caucasians. MA European patients differ from Asian subjects in timing of vasculopathy onset, lower rate of hemorrhages [5], and biphasic age distribution, with a female predominance (2.9:1, female-male ratio) more pronounced among MA adult patients [6,7].

The pathogenesis of MA is unknown. The characteristics of the stenotic change seen in MA are quite different from those of the atherosclerotic process. Indeed, there is no lipid pool or inflammatory cells or macrophage invasion to the subintimal layer as typically seen in atherosclerosis [8]. Anomalies in angiogenesis and vasculogenesis have been invoked as potential disease mechanisms due to the detection of altered levels of cytokines, chemokines, and growth factors in cerebrospinal fluid and sera of MA patients [2,9]. Conversely, the association of MA with genetic disorders, the high familial rate, and the strong association with variants of *Ring Finger Protein 213 (RNF213*) gene in East Asian patients strengthen the role of genetic factors in MA pathogenesis [10,11,12,13,14]. Overall, it is believed that MA results from a complex mechanism in which acquired infectious, inflammatory, and flow dynamic conditions may trigger the disease in genetic susceptible individuals through angiogenic and vasculogenic pathways abnormalities [2].

Circulating endothelial progenitor cells (cEPCs) are a minor population of circulating mononuclear cells derived from the bone marrow and mobilized upon specific stimulation [15]. First identified by Asahara et al. [16], cEPCs were found to be involved in both physiological and pathological postnatal vasculogenesis [17]. Once in the blood stream, cEPCs target injured tissue, promoting endothelial repair or remodeling processes [18]. Since their original identification, cEPCs have been extensively studied as biomarkers to assess the risk of cardiovascular diseases and as a potential cellular therapeutic agent for vascular regeneration [19,20]. cEPCs have been already studied in MA to better understand and characterize the disease pathogenesis, but reported results are controversial [21]. Angiographic MA vessels, characterized by a remarkable increase level of circulating cells and associated with neovascularization of the ischemic brain, have been observed in a series of MA Japanese patients [22]. An increased mobilization of peripheral blood mononuclear cells (PBMCs) has been also reported in MA European patients in comparison to healthy donors (HD) [23]. Conversely, decreased levels and defective angiogenic function of endothelial progenitor cell (EPC) were found in pediatric MA patients, possibly due to the abnormal angiogenesis during MA development [24]. An impaired function of cEPCs was also similarly observed in adult MA patients [25]. However, the results of these studies are controversial, probably due to different methodological approaches, and a consensus on the best cEPC identification/quantification approach has not been achieved yet [9,26].

The aim of the present study is to determine the level and function of cEPCs in a cohort of Italian MA patients to address the question if cEPCs may be considered a potential pathogenic marker or just an epiphenomenon of MA.

## 2. Results

### 2.1. Reduced EPC Level in Peripheral Blood of a Homogeneous Group of MA Patients

Among the original cohort of 132 patients of GEN-O-MA study [27], 47 subjects in whom it was possible to collect whole blood samples were included in the present study (Appendix A). The full study methodology has been already reported elsewhere [27]. The selected subjects displayed a mean age of 38.85 ± 17.72 years, with a prevalence of female patients (78.72%).

The disease presented with a first ischemic event in 36.17% of them, with a hemorrhagic stroke in 14.89% of them and with a transient ischemic attack (TIA) in 21.28% of cases.

Eighteen HD were recruited as controls. They displayed a mean age of 44 ± 12 years and were all females. Additionally, 10 patients (5 males and 5 females) with a mean age of 43 ± 2 years with atherosclerotic cerebrovascular disease (ACVD) were selected as further controls.

No significant differences in EPC level—identified as CD45^dim^CD34^+^CD133^+^ mononuclear cells, as detailed in the Methods section [28]—were found in MA patients in comparison with HD and ACVD controls (0.095 ± 0.197 in MA patients, 0.098 ± 0.104 in HD and 0.045 ± 0.032 in ACVD controls; *p* = 0.913 and *p* = 0.118 respectively; Figure 1A). By applying multivariate analysis, EPC levels did not correlate with age, sex, and markers of disease severity (i.e., Suzuki scale and bilateral condition; Appendix A).

To obtain a homogeneous sample from the above detailed original cohort, we selected 22 adult Caucasian MA patients who did not undergo any surgical operation, be it either direct or indirect bypass, at the time of the blood sample collection. Of these patients (mean age 46.86 ± 11.7), 86.36% were females, 27.27% presented with ischemic event, 31.81% presented with hemorrhagic stroke, and 13.64% presented with TIA. These selection criteria were applied to avoid the eventual changes in EPC recruitment that can happen following a damage to the vasculature [29].

The percentage of EPCs in this restricted group was found to be 0.035 ± 0.051 in MA patients, 0.098 ± 0.104 in HD, and 0.045 ± 0.032 in ACVD subjects. Thus, a significant decrease in the percentage of EPCs was found in MA patients compared to HD (*p =* 0.032; Figure 1B). Interestingly, EPC levels in this homogeneous group of patients correlated with Suzuki score and bilaterality (*p =* 0.026 and 0.047 respectively; Appendix A).

### 2.2. Identification and Characterization of EPCs in Culture

Since the level of the EPCs does not clarify the role of this population in the disease, other investigations were carried out on cultured EPCs. PBMCs were isolated from whole blood samples of 47 MA patients, 18 HD, and 10 ACVD subjects. From this wide and heterogeneous cell population, EPCs were identified by seeding the PBMCs in a collagen- and fibronectin-coated substrate and by growing them in Microvascular Endothelial Cell Growth Medium (EGM-MV medium), enriched with growth factors specific for endothelial cells, as reported in the literature [16].

By keeping EPCs in culture for one month, photos were taken with the Nikon Eclipse TE300 microscope at days 5, 7, 10, 17, and 31 to keep track of their morphology across the whole period of culture. As can be seen from Figure 2, the putative EPCs present themselves with a rounded shape at day 5 from the seeding in culture, acquiring within the following week the elongated, spindle-like shape typical of the early EPCs. Noticeably, only for 27 out of 47 MA patients, 13 out of 18 HD, and 3 out of 10 ACVD subjects, it was possible to successfully set up and carry out EPC cell cultures. In the other cases, the lack of PBMC adhesion in culture occurred early, preventing the collection of the corresponding samples (conditioned media and cells for RNA extraction) for subsequent analyses.

In order to establish if there is any difference in the proliferation rate and in the survival capacities between EPCs derived from HD, MA, or ACVD subjects, we have evaluated the number of cells during the EPC differentiation process by microscope cell-counting (Table 1). Specifically, we found that the values of cell number were similar in HD, MA, or ACVD- derived EPC cultures for all the examined detection times (days 7, 17, and 31).

The presence of EPCs was confirmed at day 31 from seeding when the cellular population was incubated with FITC-Ulex-lectin, commonly used to identify specific antigens on the membrane of endothelial cells [30].

To verify the presence of cells corresponding to the green signal emitted by the FITC-Ulex-lectin, a counterstaining with DAPI was performed to visualize the nuclei (Figure 3).

Through the morphological identification of EPCs, no apparent differences between MA and HD or ACVD-derived EPCs were evidenced in the differentiation process. Moreover, the putative late EPCs, morphologically identified as such and still present in culture at day 31, were confirmed to be endothelial-like cells through the identification by fluorescence, confirming that the growing conditions adopted were specific for that cellular type.

To expand the morphological and phenotypical characterization of EPCs, we have assayed the gene expression of additional endothelial differentiation markers. The mRNA expression levels of *von Willebrand Factor* (*vWF*), *CD31* and *KDR/Vascular endothelial growth factor-receptor2* (*VEGF-R2*) genes have been compared between days 17 and 31 in HD, MA, or ACVD-derived EPC cultures. Specifically, we observed an increased expression for *vWF* and *CD31* genes in all the tested conditions, with particular significance for the upregulation of *CD31* mRNA in EPCs from ACVD subjects at day 31 (*p =* 0.001) (Figure 4B). The *KDR/VEGF-R2* gene displayed a different modulation in EPCs from HD, where it was found significantly downregulated at day 31 as compared to day 17 (*p =* 0.013) (Figure 4C).

### 2.3. EPC Paracrine Activity Does Not Influence the Formation of Vessels

As reported in literature, early and late EPCs are characterized by different angiogenic properties: while the early EPCs influence angiogenesis by releasing growth factors in a paracrine way, late EPCs are thought to be the ones physically recruited in order to build new vessels [16,31,32,33].

For this reason, the effect of the conditioned media, collected from the EPC cultures at days 7 and 17, on the Human Umbilical Vein Endothelial Cell (HUVEC) capability to form vessels when seeded on a Matrigel-coated surface, which mimics the presence of the extracellular matrix of the neurovascular unit [34], was evaluated. In this assay, HUVEC cells display a propensity to migrate and create cell-cell interactions to build a vascular network, as can be seen from Figure 5. Specifically, we collected EPC-conditioned media from the heterogeneous group of MA patients (*n* = 27 out of 47 at day 7 and *n* = 20 out of 47 at day 17, respectively) and we compared them with EPC conditioned media from HD (*n* = 13 out of 18 at day 7 and *n* = 10 out of 18 at day 17, respectively). The EPC cultures reaching day 17 were in reduced numbers as compared to day 7, both for MA patients and HD.

For this functional characterization, five parameters were considered: the area covered by the HUVEC cells, the total number of branching points, the total loop number, the total tube number, and the total tube length.

To exclude the possible effect of growth factors already present in the growth medium on the tube formation capacity, the results obtained with cells growing in the conditioned media from HD and MA cultures were normalized on the HUVEC cells growing in EGM-MV medium, used as a basal control (Table 2 and Figure 6).

Conditioned media collected both from HD and MA EPC cultures did not differently influence the tube formation ability of HUVEC cells, neither at day 7 nor 17, in any of the analyzed parameters (Figure 6).

Similarly, when we collected EPC-conditioned media from the homogeneous group of MA patients (*n* = 14 at day 7 and 17, respectively) and we compared them with EPC conditioned media from HD (*n* = 13 at day 7 and *n* = 10 at day 17, respectively), we did not find any difference in tube formation capability (Table 3 and Figure 7).

### 2.4. Release of Endothelial Markers and Inflammatory Cytokines by EPCs from HD and MA Subjects

The EPC cultures were characterized by measuring the release of some endothelial markers and inflammatory cytokines in the culture media at days 7 and 17 from the seeding (Figure 8). Specifically, we assayed VEGF-A (Figure 8A), hepatocyte growth factor (HGF; Figure 8B), transforming growth factor-beta 1 (TGF-β1; Figure 8C), chemokine (C-C motif) ligand 5 (CCL5/RANTES; Figure 8D), chemokine (C-C motif) ligand 2 (CCL2/MCP-1; Figure 8E), and interleukin 8 (IL-8/CXCL8; Figure 8F).

In the conditioned media collected at day 7 from HD and MA PBMC cultures, no statistically significant difference was found for the release of VEGF-A, HGF, TGF-β1, CCL2/MCP-1, and IL-8/CXCL8 concentrations (Figure 8 and Appendix A). Interestingly, CCL5/RANTES concentration was significantly decreased in MA compared with HD, 120.57 ± 77.78 pg/mL in HD and 59.62 ± 31.59 pg/mL respectively (*p* = 0.024) (Figure 8D).

In conditioned media taken at day 17, we did not observe any significant difference in the release of endothelial markers and inflammatory cytokines (Figure 8 and Appendix A).

A further characterization of the cultured early EPCs at day 7 from seeding was carried out by evaluating the expression level of an endothelial phenotypic marker (*CD31*) and two angiogenic growth factors (*HGF* and *TGF-β1*) since their gene expression was known for being correlated to the pathology [9]. As detailed above, we successfully established EPC cultures endowed with enough cells to be assayed for RNA expression only for a limited number of cases (*n* = 7 out of 27 EPC cultures from MA patients and *n* = 4 out of 13 EPC cultures from HD at day 7).

EPCs collected from MA patients did not manifest a significantly different expression level neither for *CD31* nor *HGF*, as compared to HD (fold change 1.70 ± 1.35 and 1.13 ± 0.69; *p* = 0.25 and *p =* 0.67 respectively).

*TGF-β1* displayed instead a significant upregulation (fold change 5 ± 3.3; *p* = 0.02) in EPCs from MA patients when compared with HD (Figure 9).

## 3. Discussion

A number of evidences suggest that cEPCs contribute to the development of diverse cardiovascular diseases and to ongoing endothelial repair [35]. Reduced levels of cEPCs have been shown to be related to endothelial dysfunction, cerebral infarction, and coronary artery disease, which suggests that EPCs play an important role in vascular homeostasis [35,36,37,38]. Furthermore, it was suggested that reduced numbers and impaired functions of cEPCs are related to the pathogenesis of stroke [39,40]. In previous report, EPCs or vascular progenitor cells were shown to play an important role in physiological or pathological angiogenesis [41]. Indeed, EPCs seem to have two different roles: a structural function, as they are actively recruited at the site of new vessels growth [42], and a regulatory function, as they are source of angiogenic factors [22].

Although there are still controversial results and the exact role of EPCs has not yet been conclusively clarified, many studies have demonstrated an intimate correlation between EPCs and the development of MA (Table 4) [21,23,24]. Very recently, transcriptomic and metabolomic profiling approaches have been reported for discovering potential disease biomarkers in peripheral blood of MA patients [43,44]. Hur et al. have first in vitro characterized early and late EPCs from a single MA patient, which shared some endothelial phenotypes but showed different morphology, proliferation rate, survival features, and gene expression profiles. Interestingly, the differences observed in vitro between early and late EPCs disappeared in vivo [32]. Moreover, despite also being reported that the cEPCs in MA patients are dysfunctional, the reason for the dysfunction is poorly understood [24]. Abnormalities in EPC mitochondria have been recently related to the delayed repair of the damaged vessels and to the development of vessel occlusion in MA patients [45]. In addition, mitochondrial dysfunction-mediated decline in angiogenic capacity of EPCs is associated to capillary rarefaction in patients with hypertension [46]. 

However, the results of the studies conducted so far did not report convincing results, hampering our understanding of the role of EPCs in the MA pathophysiological mechanisms (Table 4) [7,47,48,49]. The heterogeneous results of previous studies may be due to the use of different methodological approach and to the selection bias in the evaluated patient and control cohorts, where young and adult subjects, Caucasian and Asian ethnicities, and operated vs non-operated patients were put together [9,26].

Herein, we reported the EPC level on a homogeneous MA population and we characterized the phenotypical, molecular and functional aspects of such progenitor cells.

Differently from previous reports and since EPCs should express at least one marker of immaturity and one additional marker reflecting endothelial commitment [24], we identified EPCs from whole blood as CD45^dim^CD34^+^CD133^+^ mononuclear cells [28]. The results of our study could be different from others probably due to the specific methodological approach used for cEPC identification/quantification. In fact, increasingly complex antigenic phenotypes might be more specific for EPCs but have lower reproducibility, thus limiting their use in daily clinical practice [50].

We observed a significantly reduced level of cEPCs in a homogeneous and selected group of adult, Caucasian, non-operated MA patients with respect to HD. The selection of adult MA patients and the fact that the mean ages of our groups of subjects (HD, MA, and ACVD) are well balanced were crucial points for supporting our conclusions. EPC levels seem to be related to the severity and the bilaterality of the condition, although the significance of these findings is limited by the small sample size.

The subsequent analyses carried out to characterize cultured EPCs did not show marked differences in the morphological appearance of EPCs from MA patients, HD, or ACVD subjects. In the first two weeks from culture starting, the putative EPCs acquired the elongated, spindle-like shape typical of early EPCs, with the appearance of some cobblestone-shaped cells between two and three weeks from the seeding as previously reported [32].

Moreover, EPCs from HD, MA, and ACVD subjects displayed a comparable low proliferative capacity, a common reduction in number of cells with the progression of the culture, and an expected increase in expression levels of some endothelial markers, with the exception of a decrease in gene expression for *KDR/VEGF-R2* in HD-derived EPCs. Specifically, the increase in *TGF-β1* mRNA expression in EPCs from MA patients as compared to HD is in line with a previous analysis performed on smooth muscle cells derived from the MA superior temporal artery might support the involvement of TGF-β1 in the mechanism of intimal thickening typical of the disease [59]. However, given the small sample of evaluated EPCs, it needs confirmation by further analysis. The paracrine activity of cultured EPCs did not differ between MA patients and HD. A slightly decreased CCL5/RANTES level in MA patients as compared to HD subjects was found only at an early time.

Coherently with such evidence, the cellular capacity to form vessels did not show any differences between the two conditions, thus confirming the absence of paracrine angiogenic activity of the late EPCs on the vessel formation, as previously reported [9,33,60].

Taken together, these data might suggest that, although the number of EPCs is reduced in MA patients, the proliferation ability, the gene expression profile, and the functionality of EPCs are similar in MA patients and HD subjects. Thus, a defective recruitment mechanism from bone marrow could be hypothesized in MA to explain the reduced level of EPCs evidenced in MA patients. Although our findings need confirmation in wider MA populations and further experimental models are necessary to assess whether the defective EPC recruitment has a causative role in MA pathogenesis, our results might prelude to a potential cell therapy option to cure MA conditions. So far, human endothelial colony forming cells (ECFCs) have been recognized with unique capabilities to form vascular structures in several animal models of ischemia but these cells have not been fully characterized and additional strategies are needed to improve their therapeutic potential [61,62,63]. Indeed, these highly proliferative cells possess the unique property of forming functional blood vessels in vivo upon transplantation [64,65,66,67]. This property distinguishes ECFCs from more mature endothelial cells and makes them particularly attractive as a cell therapy for enhanced vascular repair in ischemic diseases such as peripheral arterial disease, ischemic retinopathy, myocardial infarction, or stroke [68,69,70,71]. Upon transplantation at sites of ischemic injury, ECFCs incorporate into damaged blood vessels and release proangiogenic growth factors to facilitate the repair process, ultimately improving blood perfusion and organ function [61,72,73,74,75].

Moreover, both mature circulating endothelial cells (CECs), that are markers of endothelial injury, and EPCs—hallmarks of regeneration—express the adhesion molecule CD146 [76]. In patients with myocardial infarction, alterations of these two subsets may attest to the extent of vascular injury and the associated angiogenic response [76]. Other results suggested that endothelial injury, observed as an increased CEC number, and impaired regeneration, reflected by a lowered CEPCs/CECs ratio, preceded left ventricular hypertrophy in hypertensive patient’s occurrence [77].

While ECFCs, CECs, or EPCs represent promising candidates for cell therapy, it is critical to improve their efficacy in vivo through promoting cell survival and increasing the kinetics of cell migration, homing, differentiation and secretion of proangiogenic factors [63,68,78]. In this context, an important role of coexisting microenvironment has to be also considered because an insufficient production of IL-10 from monocytes/macrophages has been recently suggested as the mechanism that impairs in vitro EPC differentiation from cultured PBMCs of MA patients [58]. Interestingly, autologous bone marrow stem cell mobilization combined with dexamethasone treatment has been recently proposed as a therapeutic option after revascularization in MA patients [79].

There are several limitations to our study: firstly, the sample size of MA patients and controls is relatively small, mostly after patient selection. Thus, our findings should be verified in a larger cohort. Moreover, the study could suffer from selection bias since a strict sex- and age-matched control matching has not been fully respected, particularly in the relatively small group of ACVD subjects. Lastly, given the practical and ethical difficulty of acquiring the pathological tissue of MA patients, cultured EPCs were used for the present study in the absence of a more realistic disease cellular model. In this regard, only for a limited number of MA patients, HD, and ACVD subjects it was possible to successfully set up and carry out EPC cell cultures able to be assayed for subsequent molecular analyses. The variability in isolation of specific subpopulations of cells, the dynamic nature of cell marker expression, and the lack of standardization in measures of cell function are technical issues that currently limit the utility of this modeling approach. For example, the relevance of CD146 as a marker for the identification of a subpopulation of cEPC with particular propensity for a potential proangiogenic therapy and for the monitoring of vascular injury/regeneration processes in selected clinical conditions (i.e., myocardial infarction) has been suggested [76]. However, increased complexity of the antigenic combination, despite providing additional information about the cells under investigation, does not necessarily improve the performance of the cells as clinical biomarkers. Indeed, a disease biomarker does not necessarily have to be highly biologically informative, but it is required to have strong statistical associations with several clinical aspects of the disease [50]. Nevertheless, since cytofluorimetric analysis allows a more confident identification of cells on the base of their molecular asset, we prospect to confirm the obtained results by using vital antibodies at different stages of EPC maturation to further identify the EPCs in culture and to distinguish between early and late EPCs.

## 4. Materials and Methods 

### 4.1. Moyamoya Patients and Healthy/ACVD Controls: Inclusion Criteria

This was an observational study conducted on MA patients, diagnosed following the literature criteria [80], belonging to the GEN-O-MA study. The full methodology of the study is reported elsewhere [27]. From the original population of 132 patients consecutively enrolled at the Neurology IX Unit of the Fondazione IRCCS Istituto Neurologico “C. Besta” (Milan), between November 2014 and November 2019, a patient subgroup was selected for the present study (see Appendix A for clinical-demographical characteristics). Individuals fasted within 12 h, and subjects with endometriosis and/or positive for HIV, HBV, or HCV were excluded from this study.

A population of age- and sex- matched Caucasian healthy donors (HD) recruited from the general population has been collected as control. Since vascular risk factors could influence cEPC populations, HD were interviewed for their medical history, and subjects presenting at least one of the following parameters out of normality ranges were excluded: blood pressure, glycemia, and cholesterol level [81,82,83,84]. Smokers; subjects with active duodenal or gastric ulcer; subjects that had undergone drug treatments in the preceding 48 h; or subjects with present or previous neoplastic, infectious, inflammatory, or cardiovascular diseases have been also excluded.

Additionally, a group of age- and sex- matched Caucasian atherosclerotic cerebrovascular disease (ACVD) patients were recruited as further controls. ACVD was diagnosed when patients had either an internal carotid, middle cerebral artery, cerebral anterior artery occlusion or stenosis from atherosclerotic origin. ACVD patients underwent conventional catheter digital subtraction angiography and morphological imaging by MRI as well.

### 4.2. Ethical Issues

The study design was approved by the Ethics Committee of the Fondazione IRCCS Istituto Neurologico “C. Besta” of Milan (report no. 12, 10/01/2014) and was performed in accordance with the 2013 WMA Declaration of Helsinki. Since it was designed as a pure observational study, patients underwent to diagnostic procedures and received therapy according to local practice. Informed written consent for study participation and samples collection from all patients and controls were mandatory for study inclusion. Privacy procedures were applied to protect patients’ and healthy controls’ personal identities.

### 4.3. Blood and Plasma Samples Collection

Twenty-four (24) microliters of peripheral blood were withdrawn by venipuncture from MA, HD, and ACVD subjects and collected in tubes containing ethylenediaminetetraacetic acid (EDTA) as anticoagulants (Vacuette^®^, Preanalitica s.r.l., Caravaggio, Italy).

One vacutainer was stored at −20 °C for future molecular analysis. For plasma collection, two vacutainers were centrifuged for 10 min at 300× *g*; plasma was transferred into a new tube (SARSTEDT AG & Co, Nümbrecht, Germany) and stored in aliquots at −80 °C until use.

### 4.4. Clinical-Radiological Factors

For all patients, demographic and clinical features were collected applying a standardized form [27].

MA was classified into the bilateral or unilateral types depending on the number of distal ICAs involved, as observed on conventional angiography [80]. Diagnosis of ischemic or hemorrhagic stroke was confirmed by conventional neuroimaging (computerized tomography scan and magnetic resonance imaging). The MA severity was assessed by Suzuki scale [85].

### 4.5. Flow Cytometry Analysis 

Flow cytometry analysis of cEPCs was performed on fresh whole blood using Flow-Count Fluorospheres (Beckman Coulter s.r.l., Cassina De’ Pecchi, Italy). Fifty microliters of whole blood (EDTA), mixed with equal volume of Flow-Count Fluorospheres (Beckman Coulter s.r.l., Brea, CA, USA), were incubated with 10 µL of monoclonal antibodies anti-CD45-Pe vio770, anti-CD34-FITC, and anti-CD133-PE (Miltenyi Biotech, Bergisch Gladbach, Germany) for 30 min at +4 °C in the dark. Flow-Count Fluorospheres are a suspension of fluorescent microbeads used to determine absolute counts on the flow cytometer. Each fluorosphere contains a dye which has a fluorescent emission range of 525 nm to 700 nm when excited at 488 nm. They have uniform size and fluorescence intensity and an assayed concentration allowing a direct determination of absolute counts. They have been used as an aid in optimizing a quantitative Fluorescence-activated cell sorting (FACS) analysis. Then, erythrocytes were lysed and leucocytes were fixed with Uti-Lyse kit (DakoCytomation, Glostrup, Denmark). Aspecific staining was determined with appropriate Isotype Control (BD Bioscience, San Jose, CA, USA), and samples were analyzed within a week of blood collection in a FACSCalibur flow cytometer (BD Bioscience, San Jose, CA, USA) equipped with CellQuest software. 

We considered as cEPCs CD45^dim^CD34^+^CD133^+^ mononuclear cells. The cellular population selection criteria based on a previous paper by our institution, although referring to a different disease, allowed us to fine-tune the setting of the flow cytometry parameters by creating gates to distinguish the EPC cellular component of interest in the HD group (Appendix A) [28]. Since cEPCs are rare in normal peripheral blood, at least 500 CD34^+^ cells per sample were acquired and nonviable cells were excluded by physical gating. For sample normalization, a complete white blood cell count was performed by the cell counter Advia 120 (Bayer, Leverkusen, Germany). 

The percentage of cEPCs was calculated as follows: % cEPCs = (cEPCs/µl / WBC/µl) × 100.

### 4.6. Isolation and Culture of Endothelial Progenitor Cells

EPCs were cultivated as previously described by Asahara T. and colleagues [16]. PBMCs were separated by Ficoll density-gradient centrifugation (Ficoll-Paque^TM^ Plus, GE Healthcare, Freiburg Germany). Then, the isolated PBMCs were suspended in Endothelial Cell Basal Medium (EBM medium, Lonza, Walkersville, MD, USA) supplemented with 10% fetal bovine serum (Gibco^®^ Life Technologies, ThermoFischer, Monza, Italy), 2 mM glutamine (Euroclone, Pero, Italy), and EGM-MV bullet kit (Microvascular Endothelial Cell Growth Medium SingleQuotsTM Supplements, Lonza, Walkersville, MD, USA) and seeded at the density of 5 × 10^6^ in 1 μg/cm^2^ collagen Bornstein and Traub type IV from human placenta (SigmaAldrich Co., St. Louis, MO, USA) and 1 µg/cm^2^ bovine plasma fibronectin (Invitrogen, ThermoFischer, Monza, Italy, P/M 42805)-coated 6-well plates. Cells were incubated at 37 °C and 5% CO_2_.

Starting from the third day after seeding, the medium was changed every two days.

Conditioned media were collected at 7 and 17 days after seeding. They were centrifuged at 300× *g* for 10 min and stored at −80 °C until use.

In order to monitor EPC morphology within 31 days of culture, at 5, 10, 17, and 31 days after seeding, photos were taken with a Nikon Eclipse TE300 microscope (Nikon Instruments Europe BV, Amsterdam, The Netherlands) with the camera Axiovision device (Zeiss Instr., Gollingen, Germany) at the magnification of 20× and images were processed with the software Axiovision release 4.6.3 (Zeiss Instr., Gollingen, Germany).

### 4.7. Fluorescence Microscopy 

The presence of EPCs in culture from MA peripheral blood samples was verified by staining the cells with FITC-lectin from Ulex Europaeus (gorse, furze) (SigmaAldrich Co., St. Louis, MO, USA), an endothelial-like cell marker.

At 31 days after seeding, the cell growth medium was discarded and the cells were washed with PBS. After fixation in 4% paraformaldehyde (MERCK, Darmstadt, Germany) in PBS for 30 min at room temperature, cells were incubated overnight at 4 °C with FITC-Ulex-lectin (1:100). Then, cells were washed 4 times in PBS, permeabilized with PBS-Tween 1% (BIORAD, Foster City, CA, USA) for 10 min, incubated with DAPI (1:1000) (SigmaAldrich Co., St. Louis, MO, USA) for 10 min at room temperature, and washed 4 times with PBS. Photos were taken by a Nikon Eclipse TE300 microscope with the camera Axiovision device (Zeiss Instr., Gollingen, Germany) at the magnification of 20×, and images were processed with the software Axiovision release 4.6.3 (Zeiss Instr., Gollingen, Germany).

### 4.8. Cell Culture

Human Umbilical Vein Endothelial Cells (HUVECs) were obtained from ATCC (ATCC^®^ CRL-1730™) and maintained in Endothelial Cell Basal Medium-2 (EBM-2 medium, Lonza, Walkersville, MD, USA) supplemented with 10% fetal bovine serum (Gibco^®^ Life Technologies, ThermoFischer, Monza, Italy) and EGM-2 bullet kit (Microvascular Endothelial Cell growth Medium SingleQuots^TM^ supplements, Lonza, Walkersville, MD, USA) at 37 °C and 5% CO_2_. HUVECs were seeded in plates coated with 1 μg/cm^2^ collagen Bornstein and Traub type IV from human placenta (SigmaAldrich Co., St. Louis, MO, USA). 

### 4.9. Tube Formation Assay

The angiogenic activity of the released factors by HD and MA EPCs during their growth was assessed using the in vitro Matrigel tube formation assay.

Ninety-six-well plates were coated with 50 µL Matrigel (Matrigel growth factor reduced, Corning, Tewksbury, MA, USA) and incubated 2 h at 37 °C to polymerize. Then, HUVECs were seeded at the density of 25,000 cells/well in 100 μL of their growth medium (EGM-2 medium) and incubated 1 h at 37 °C and 5% CO_2_ to allow their adhesion. Then, the culture medium was removed and replaced with the conditioned media collected from HD and MA EPCs at 7 and 17 days after seeding. For the control condition, we used EGM-MV medium and each experiment was performed in triplicate. The plate was incubated at 37 °C and 5% CO_2_ for 16 h, and then, one photo/well was acquired with Nikon Eclipse TE300 microscope (Nikon Instruments Europe BV, Amsterdam, The Netherlands) with the camera Axiovision device (Zeiss Instr., Gollingen, Germany), using 5× magnification.

The analysis of the images was made with Wimasis software, available online at the following link: https://www.wimasis.com/en/WimTube. The output generated by the software includes information about cell covered area (%), total branching points, total loops, total tubes, and total tubes length.

### 4.10. ELISA

VEGF-A (PEPROTECH, Rocky Hill, NJ, USA), HGF (R&D Systems, Minneapolis, MN, USA), TGF-β1 (R&D Systems, Minneapolis, MN, USA), IL-8/CXCL8 (ThermoFisher, Monza, Italy), CCL2/MCP-1 (Diaclone, Besancon, France), and CCL5/RANTES (ThermoFisher, Monza, Italy) concentrations were assessed using highly sensitive enzyme-linked immunosorbent assay kit in triplicate samples obtained from EPC conditioned media. Enzyme-linked immunosorbent assay was performed according to the manufacturer’s instructions. 

### 4.11. RNA Extraction and Real-Time PCR Analysis 

Total RNA was extracted from EPCs at 7, 17, and 31 days after seeding using RNeasy lipid Tissue Mini Kit (Qiagen, Valencia, CA, USA) and quantified by Qubit^®^ RNA HS assay kit (Life Technologies, ThermoFischer, Monza, Italy), and then 1 µg RNA was reverse-transcribed with iScript Advanced cDNA Synthesis Kit (BIORAD, Hercules, CA, USA), according to the manufacturer’s protocol; 2.5 μL cDNA was amplified by CFX-96 Real Time PCR Detection System (BIORAD, Foster City, CA, USA) using iTaqTM Universal SYBR^®^ Green SuperMix (BIORAD, Hercules, CA, USA). The relative level of *HGF*, *CD31*, *TGF-β1*, *vWF*, and *KDR* mRNA was calculated by the 2^−ΔΔCt^ comparative method using *β2M* as the housekeeping gene. EPCs from HD were chosen as the calibrator. Primer sequences were designed using Primer3 free online software (Simgene.com). The primer sequences are *vWF*: forward: ATGAGTGTGCCTGCAACTGT; reverse: GACACACACCTTGTCGGGAA; *KDR*: forward: GAGGGGAACTGAAGACAGGC; reverse: GGCCAAGAGGCTTACCTAGC; *HGF*: forward ACGAACACAGCTTTTTGCCTT, reverse CCCCTCGAGGATTTCGACAG; *CD31*: forward GTGTCTTGAGTGGGTGGGAG, reverse AGGCGTGGTTCTCATCTGTG; *TGF-β1*: forward ACCTGCCACAGATCCCCTAT, reverse CTCCCGGCAAAAGGTAGGAG; and *β2M*: forward CTGCCGTGTGAACCATGTGA, reverse CTTCAAACCTCCATGATGCTGC. The analysis was performed with BIORAD CFX Manager software (BIORAD, Foster City, CA, USA).

### 4.12. Statistical Analysis

Data were expressed as mean ± standard deviation, and statistical significance (* *p* < 0.05, ** *p* < 0.01, and *** *p* < 0.001) was calculated through Student’s *t*-test by using GraphPad Prism 8 software. Multivariate regression analysis adjusted for age and sex was applied to assess the independence of disease severity features. EPC levels were used as the dependent variable for each single feature. 

Tests were considered significant if the *p* value was <0.05. All analyses were performed using STATA 8.0 software.

## 5. Conclusions

It is increasingly evident that the pathophysiology of MA encompasses many different mechanisms and that the use of biomarkers will refine and improve our understanding of this arteriopathy. In this respect, circulating EPCs may provide a useful combination of molecular and functional markers of the disease.

Therefore, the variety of methods used to study EPCs in published clinical studies is so wide that specification of the exact method used in each work becomes critical. Indeed, even when looking at the same clinical condition, studies using different methodologies could come to opposite conclusions.

Since MA manifests with high heterogeneity, a careful selection of patients based on their clinical/neuroradiological profile suggests that a reduction in circulating EPC level could represent a potential pathogenic marker of MA. The validation of our results on a larger population and the correlation with clinical data could help our understanding of EPC role in MA.

Future efforts will benefit from multicenter studies and working groups in guiding utilization in clinical practice.

## Figures and Tables

**Figure 1 ijms-21-05763-f001:**
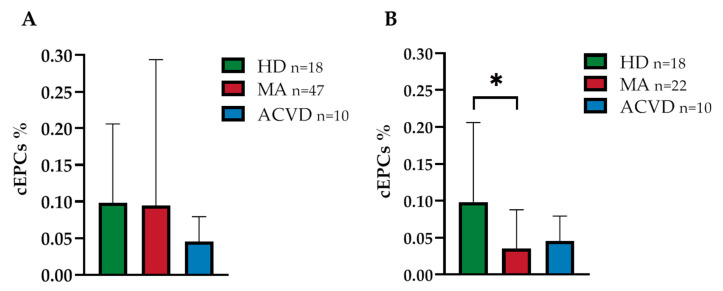
Endothelial progenitor cell (EPC) level in whole blood (WB) from (**A**) a heterogeneous and (**B**) a homogeneous (adult, Caucasian, non-operated) group of Moyamoya angiopathy (MA) patients, as compared with healthy donors (HD) and atherosclerotic cerebrovascular disease (ACVD) controls: Data are expressed as mean of circulating endothelial progenitor cell (cEPC)% ± standard deviation (SD), where cEPC% was calculated as follows: (cEPCs/µL/WB cells/μL) × 100; statistical significance (* *p* < 0.05) was calculated through Student’s *t*-test (*p =* 0.032).

**Figure 2 ijms-21-05763-f002:**
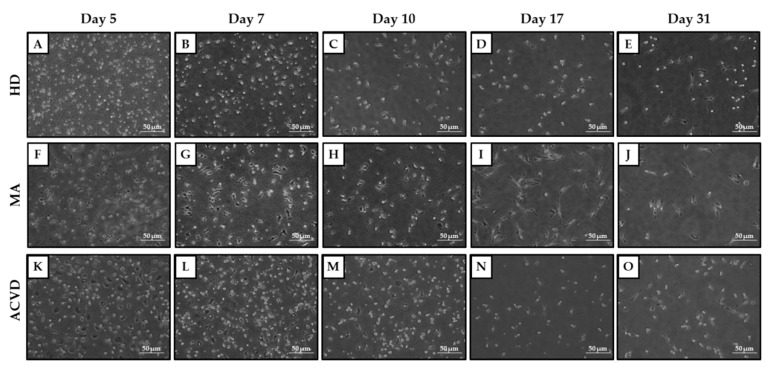
Cultured EPCs from (**A**–**E**) HD, (**F**–**J**) MA, and (**K**–**O**) ACVD representative subjects at 5, 7, 10, 17, and 31 days after seeding in Microvascular Endothelial Cell Growth Medium (EGM-MV medium) (20× magnification, 50 μm scale bar).

**Figure 3 ijms-21-05763-f003:**
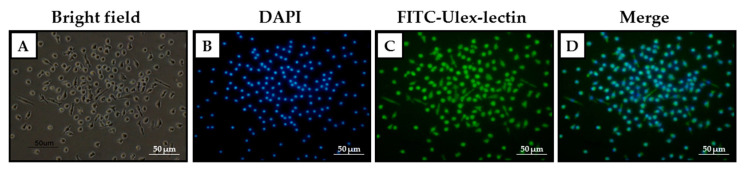
Staining of late EPCs obtained from a representative MA patient by (**A**) Bright field and (**B**–**D**) fluorescence microscopy; (**B**) DAPI, (**C**) FITC-Ulex-lectin, (**D**) DAPI+FITC-Ulex-lectin (20× magnification, 50 μm scale bar).

**Figure 4 ijms-21-05763-f004:**
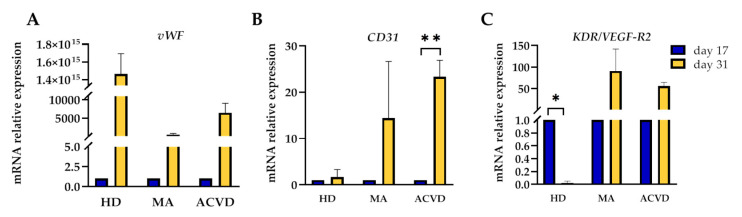
mRNA relative expression of (**A**) *von Willebrand Factor* (*vWF)*, (**B**) *CD31*, and (**C**) *KDR/ Vascular endothelial growth factor-receptor2* (*VEGF-R2)* in EPCs at 17 and 31 days after seeding: the mRNA levels in EPCs at day 31 from HD, MA, and ACVD subjects are expressed in relation to EPCs at day 17, arbitrarily imposed at 1 as calibrator. *β2-microglobulin* (*β2M*) was used as housekeeping gene. Data were expressed as mean ± SD, and statistical significance (* *p* < 0.05, *** p* < 0.01) was calculated through Student’s *t*-test. Values of at least three independent experiments are shown.

**Figure 5 ijms-21-05763-f005:**
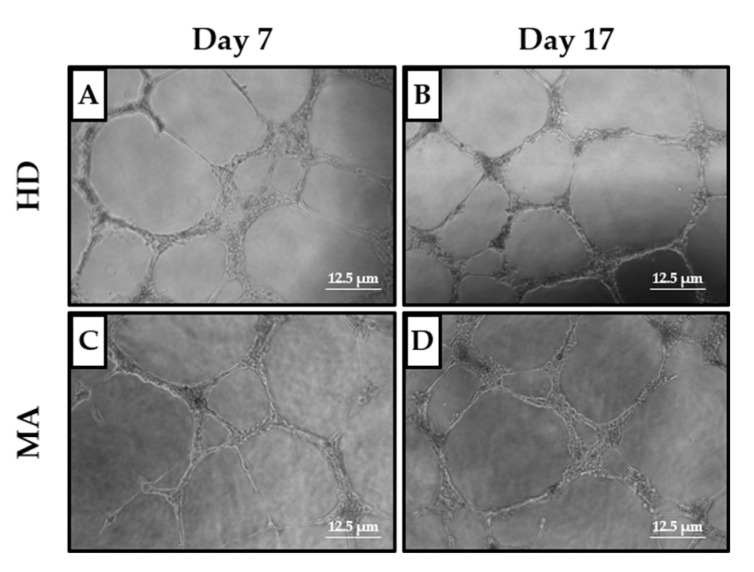
Tube formation assay on Human Umbilical Vein Endothelial Cell (HUVEC) cells in the presence of conditioned media from HD and MA EPC cultures at (**A**,**C**) day 7 and (**B**,**D**) day 17 (representative images are shown; 5× magnification, 12.5 μm scale bar): The angiogenic potential of the growth factors released by early and late EPCs was evaluated through the analysis of the photos from each condition by the Wimasis software. Three independent experiments for each condition have been considered.

**Figure 6 ijms-21-05763-f006:**
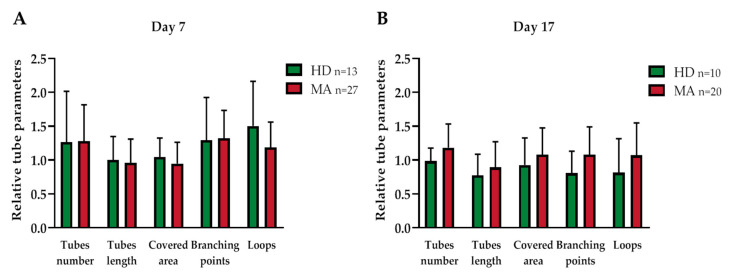
Vasculogenic capacity of HUVEC cells cultured in conditioned media collected (**A**) 7 and (**B**) 17 days after seeding of EPCs from HD and from a heterogeneous group of MA patients (each different tube parameter has been normalized with values obtained by HUVEC cells in EGM-MV medium): Data were expressed as mean ± SD, and statistical significance was calculated through Student’s *t*-test. Values of at least three independent experiments are shown.

**Figure 7 ijms-21-05763-f007:**
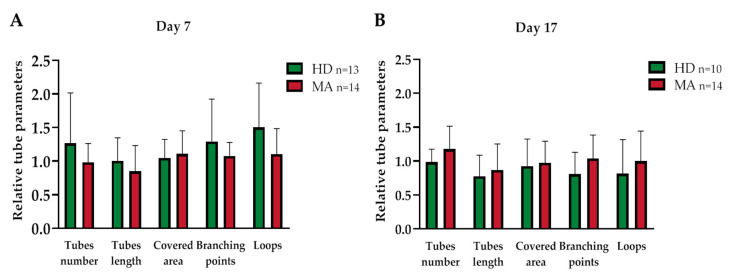
Vasculogenic capacity of HUVEC cells cultured in conditioned media collected (**A**) 7 and (**B**) 17 days after seeding of EPCs from HD and from a homogeneous group of MA patients (each different tube parameter has been normalized with values obtained by HUVEC cells in EGM-MV medium): Data were expressed as mean ± SD, and statistical significance was calculated through Student’s *t*-test. Values of at least three independent experiments are shown.

**Figure 8 ijms-21-05763-f008:**
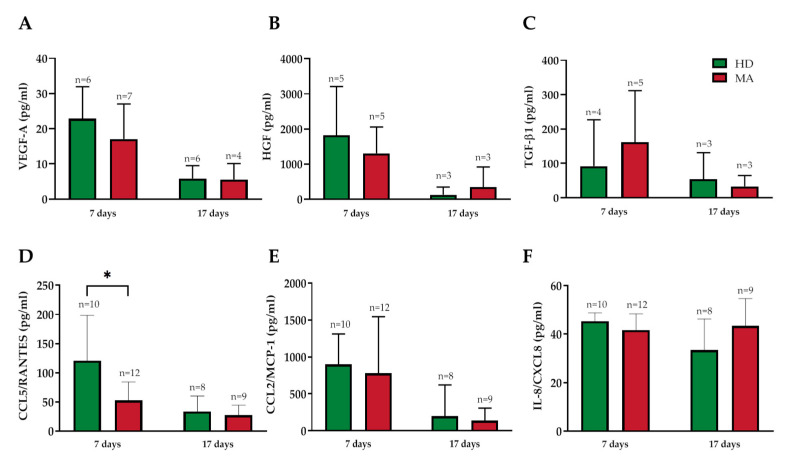
(**A**) Vascular endothelial growth factor A (VEGF-A), (**B**) hepatocyte growth factor (HGF), (**C**) transforming growth factor-beta 1 (TGF-β1), (**D**) chemokine (C-C motif) ligand 5 (CCL5/RANTES), (**E**) chemokine (C-C motif) ligand 2 (CCL2/MCP-1), and (**F**) interleukin 8 (IL-8/CXCL8) concentration (pg/mL) in conditioned media collected from EPC cultures at 7 and 17 days after seeding: Data were expressed as mean ± SD, and statistical significance (* *p* < 0.05) was calculated through Student’s *t*-test. Values of at least three independent experiments are shown.

**Figure 9 ijms-21-05763-f009:**
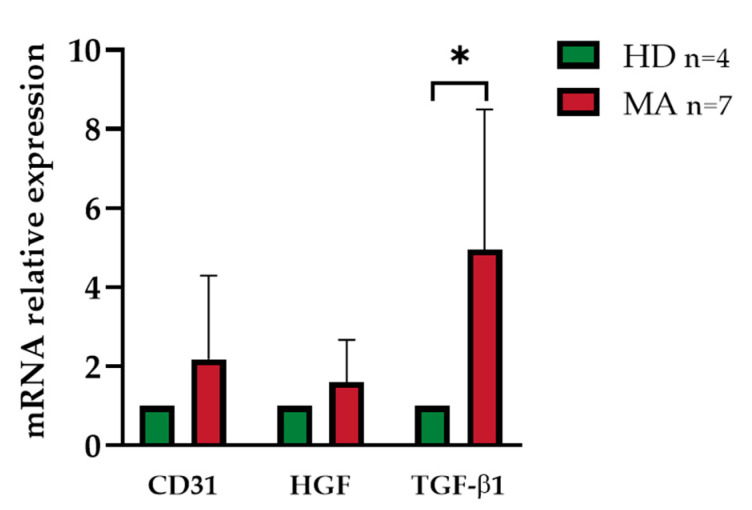
mRNA relative expression of *CD31*, *HGF*, and *TGF-β1* in EPCs at 7 days after seeding: The mRNA levels in EPC from MA patients are expressed in relation to HD subjects, arbitrarily imposed at 1 as calibrator. *β2M* was used as housekeeping gene. Data were expressed as mean ± SD, and statistical significance (* *p* < 0.05) was calculated through Student’s *t*-test. Values of at least three independent experiments are shown.

**Table 1 ijms-21-05763-t001:** Number of cells measured at days 7, 17, and 31 after seeding of EPCs by HD, MA, and ACVD subjects: Data were expressed as mean ± SD, and statistical significance was calculated through Student’s *t*-test by comparing MA and ACVD values to HD ones. EPC numbers, detected by counting four culture wells of at least three independent experiments, are shown.

Subjects	7 days	*p* Value	17 days	*p* Value	31 days	*p* Value
**HD**	26.75 ± 16.97		9.25 ± 2.28		6.75 ± 2.28	
**MA**	28.89 ± 30.33	0.647	10.78 ± 13.46	0.681	5.5 ± 8.07	0.603
**ACVD**	19.5 ± 4.5	0.187	11 ± 5	0.787	5.5 ± 5.5	0.858

**Table 2 ijms-21-05763-t002:** Normalized values for % of covered area, number of branching points, total number of loops, total tube number, and total tube lengths in the tube formation assay, performed with conditioned media collected at days 7 and 17 after seeding of EPCs of HD and of a heterogeneous group of MA patients: Data were expressed as mean ± SD, and statistical significance was calculated through Student’s *t*-test. Values of at least three independent experiments are shown.

Day 7				Day 17		
Parameters	HD	MA	*p* Value	HD	MA	*p* Value
*n*	13	27		10	20	
Covered area	1.045 ± 0.268	0.947 ± 0.309	0.329	0.926 ± 0.380	1.080 ± 0.383	0.330
Branching points	1.293 ± 0.606	1.323 ± 0.403	0.881	0.808 ± 0.306	1.081 ± 0.399	0.059
Total loops	1.502 ± 0.636	1.187 ± 0.368	0.13	0.813 ± 0.477	1.07 ± 0.467	0.198
Total tubes	1.265 ± 0.722	1.278 ± 0.528	0.956	0.985 ± 0.181	1.182 ± 0.342	0.056
Total tube length	1.004 ± 0.331	0.957 ± 0.346	0.691	0.774 ± 0.296	0.892 ± 0.37	0.375

**Table 3 ijms-21-05763-t003:** Normalized values for % of covered area, number of branching points, total number of loops, total tube number, and total tube lengths in the tube formation assay, performed with conditioned media collected at days 7 and 17 after seeding of EPCs of HD and of a homogeneous group of MA patients: Data were expressed as mean ± SD, and statistical significance was calculated through Student’s *t*-test. Values of at least three independent experiments are shown.

Day 7				Day 17		
Parameters	HD	MA	*p* Value	HD	MA	*p* Value
*n*	13	14		10	14	
Covered area	1.045 ± 0.268	1.108 ± 0.331	0.602	0.926 ± 0.380	0.974 ± 0.308	0.757
Branching points	1.293 ± 0.606	1.079 ± 0.192	0.259	0.808 ± 0.306	1.037 ± 0.335	0.113
Total loops	1.502 ± 0.636	1.103 ± 0.367	0.073	0.813 ± 0.477	0.998 ± 0.429	0.364
Total tubes	1.265 ± 0.722	0.983 ± 0.271	0.222	0.985 ± 0.181	1.175 ± 0.325	0.094
Total tube length	1.004 ± 0.331	0.849 ± 0.369	0.279	0.774 ± 0.296	0.868 ± 0.37	0.516

**Table 4 ijms-21-05763-t004:** Summary of studies about EPC level in MA: Number (*n*), age (A, adult; P, pediatric), percentage of female, ethnicity, percentage of non-operated MA patients, methods for EPC identification/quantification (FACS, Fluorescence-activated cell sorting), and EPC amount variation in MA patients as compared to HD are reported (↑, EPC increased level; ↓, EPC decreased level; =, EPC unaffected level; na, not available).

Reference	*n*	Age	Female (%)	Ethnicity	Prior Surgical Treatment (%)	EPC Identification	EPC Level
[25]	24	A	66.67%	na	87.5%	Colony counting	↓
[22]	4	A	50%	na	na	FACS	↑
[23]	20	A/P	70%	na	100%	FACS	↑
[24]	28	P	50%	na	100%	FACS; colony counting	↓; ↓
[51]	18	A	50%	na	100%	FACS	↑
[52]	17	A/P	68%	na	100%	FACS	=
[53]	6	A/P	50%	na	na	FACS; colony counting	=; ↓
[54]	12	A/P	75%	na	na	FACS	=
[55]	66	A	43.94%	na	100%	FACS	=
[56]	5	P	40%	na	na	FACS	=
[57]	4	P	50%	na	na	FACS	=
[58]	23 7	A A	52.17% 71.42%	na na	100% 0%	Colony counting Colony counting	↓ ↓
Present study	22	A	86.36%	Caucasian	100%	FACS	↓

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
