# Peer review of "Vascular Remodeling in Moyamoya Angiopathy: From Peripheral Blood Mononuclear Cells to Endothelial Cells"

_ijms, 2020, doi:10.3390/ijms21165763_

Round 1

Reviewer 1 Report

Manuscript ID: ijms-871534

Title: Vascular Remodeling in Moyamoya Angiopathy: from Peripheral Blood Mononuclear Cells to Endothelial Cells. Francesca Tinelli , Sara Nava, Francesco Arioli , Gloria Bedini , Emma Scelzo , Daniela Lisini , Giuseppe Faragò , Andrea Gioppo , Elisa F. Ciceri , Francesco Acerbi , Paolo Ferroli , Ignazio G. Vetrano , Silvia Esposito, Veronica Saletti, Chiara Pantaleoni, Federica Zibordi, NardoNardocci, Maria Luisa Zedde, Alessandro Pezzini, Vincenzo Di Lazzaro, Fioravante Capone, Maria Luisa Dell’Acqua, Peter Vajkoczy, Elizabeth Tournier-Lasserve, Eugenio A. Parati, Anna Bersano, and Laura Gatti.

In this original article entitled “Vascular Remodeling in Moyamoya Angiopathy: from Peripheral Blood Mononuclear Cells to Endothelial Cells” Tinelli and colleagues showed that cEPC level, more than EPC functionality, seems to be a potential marker of MA. For that issue they analyzed the cEPC cells in the peripheral blood of Caucasian MA non-surgery patient vs healthy and ACVD patient. Moreover, they cultured EPC cells from these same patients and evaluated their behaviour and compared it with HD and ACVD.

This is an interested and well-written manuscript. However, the numbers of positive results are few, and provide little new knowledge and novelty to the field.

MAJOR POINTS

1) There are many previous studies that have studied cEPCs to better understand and characterize the disease pathogenesis. However, the controversial results about EPC in MA disease speculate about the involvement of these cells in this pathology and about the heterogeneity of the patient samples used.

In this study, the authors showed that in a homogeneous cohort of Caucasian MA patients without surgical operation the level of CD45dimCD34+CD133+ is lower comparing with ACDV and HD. In this sense, they used new markers to identify EPCs than others reports so it would be necessary to show FACS analysis plots and point out the population selection criteria.

Authors explain that differences in the % of EPC in MA patient between studies could be due to differences in methodological approach and in the selection of the patient cohort

Moreover, they described that ACVD patients do not even show differences in % of cEPCs vs HD when in previous reports (Stroke 2009, 40,432-438.doi:10.1161/STROKEAHA.108.529420) is described significant differences. Could you explain this phenomenon? . Moreover, better description of the ACVD group of patients should be done.

Authors suggested that EPC cells seem to be a potential marker for MA disease. Is there a correlation between Suzuki scope and % of cEPCs in MA patient?

2) Figure 2-3. Images, especially at 17 and 31 days display that HD and ACVD EPC cells present more rounded and death cells that MA EPCs culture. Is there any difference in the proliferation rate and in the survival capacities between cells? A cell cycle assay, number of cells or proliferation assay should be assessed.

Moreover, author described,“Through the morphological identification of EPCs, no apparent differences between MA and HD or ACVD-derived EPCs were evidenced in the differentiation process”, however to demonstrate no differences in differentiation process more endothelial markers may be assessed.

3) Figure 7-8. Authors showed that levels of some secreted molecules in supernatant of EPCs cultures do not changes to demonstrate no paracrine activity differences between HD and MA. However, detection of altered levels of cytokine and chemokines in sera of MA patients has been described (Curr. Med. Chem. 2016, 23, 315-345. doi: 546 10.2174/092986732304160204181543). Is there any difference in inflammatory cytokines release (example CCL5, CCL2, MCP-1...) in the supernatant in this case?.

Reviewer 2 Report

1. cEPC has been identified as CD45dimCD34 + CD133 + mononuclear cells, is this population accurate? The authors did not decide on CD146 (+), did this not affect the quality of the research?

2.To fully assess endothelial dysfunction in patients with MA, authors should also assess the amount of CEC followed by the cEPC / CEC ratio. The final number of EPCs seems insufficient for endothelial regeneration and organ repair, given the circulation, the amount of CEPC is regulated by other mechanisms associated with their life expectancy and death, such as apoptosis. Increased caspase-3 activity in CEPCs isolated from patients with aortic stenosis indicated that increased apoptosis is a significant cause of reduced CEPC levels. As a result, the total number of CEPCs observed in circulation is not increasing and may even fall despite continuous release from the bone marrow. Analysis of the cEPC / CEC ratio would allow assessment of imbalance between regenerative and degenerative endothelial processes.

3. Authors should expand statistics by Multivariate analysis of the relationship between the number of EPCs, clinical parameters and neurological features.

Round 2

Reviewer 1 Report

Minor comments

lines 565-568 does not appear in the main text